# Mesenchymal Stem Cells for Mitigating Radiotherapy Side Effects

**DOI:** 10.3390/cells10020294

**Published:** 2021-02-01

**Authors:** Kai-Xuan Wang, Wen-Wen Cui, Xu Yang, Ai-Bin Tao, Ting Lan, Tao-Sheng Li, Lan Luo

**Affiliations:** 1School of Medical Technology, Xuzhou Key Laboratory of Laboratory Diagnostics, Xuzhou Medical University, Xuzhou 221004, China; 301910411466@stu.xzhmu.edu.cn (K.-X.W.); 301910411469@stu.xzhmu.edu.cn (W.-W.C.); 300104120621@stu.xzhmu.edu.cn (X.Y.); tinglan@xzhmu.edu.cn (T.L.); 2Division of Cardiology, The Affiliated People’s Hospital of Jiangsu University, Zhenjiang 212132, China; taoab@jskfhn.org.cn; 3Department of Stem Cell Biology, Atomic Bomb Disease Institute, Nagasaki University, Nagasaki 852-8523, Japan

**Keywords:** radiation-induced injury, radiotherapy, mesenchymal stem cells

## Abstract

Radiation therapy for cancers also damages healthy cells and causes side effects. Depending on the dosage and exposure region, radiotherapy may induce severe and irreversible injuries to various tissues or organs, especially the skin, intestine, brain, lung, liver, and heart. Therefore, promising treatment strategies to mitigate radiation injury is in pressing need. Recently, stem cell-based therapy generates great attention in clinical care. Among these, mesenchymal stem cells are extensively applied because it is easy to access and capable of mesodermal differentiation, immunomodulation, and paracrine secretion. Here, we summarize the current attempts and discuss the future perspectives about mesenchymal stem cells (MSCs) for mitigating radiotherapy side effects.

## 1. Introduction

Malignant tumors are one of the most aggressive diseases and have high mortality. Currently, there are no efficient methods capable of eradicating cancers clinically. As a conventional cancer treatment modality, radiotherapy (RT) can kill cancer cells and improve patient survival rates. Unfortunately, cancer patients also have to risk radiotoxicity to healthy tissues around the tumor. Clinical studies have revealed skin, intestinal, brain, pulmonary, hepatic, and cardiovascular injuries in cancer patients who received RT [1,2,3,4,5]. Although developments in RT devices and techniques (e.g., intensity-modulated RT, IMRT; image-guided RT, IGRT.) have significantly decreased radiation dose, exposure volume, and area, radiation injury is still unavoidable [6,7,8,9]. There is no evidence showing the existing dose threshold that would not damage the cell [10]. Emerging epidemiological data have consistently confirmed that low-dose radiation could also cause tissue damage [11,12]. Thus, when optimizing the RT technique to reduce the risk of radiation exposure, more effort should be made to seek satisfactory treatment for radiation-induced tissue injury.

In recent decades, stem cells have become a hot topic of research in regenerative medicine, bioengineering, and other clinical settings. Among the various stem cell types, mesenchymal stem cells (MSCs) are the most frequently studied. Thousands of publications are issued, and more than 490 clinical trials utilizing MSCs have been carried out or ongoing [13]. The reasons might be that MSCs are easy to access due to their abundant resources, including bone marrow, adipose tissue, umbilical cord, and placental tissue. Additionally, MSCs possess stable genomes, great self-renewal ability, mesodermal differentiation capacity, and immunomodulatory and paracrine secretome [14]. Indeed, MSCs reveal the tremendous therapeutic potential in various diseases such as cancer, diabetes mellitus, autoimmune disease, liver injury, and cardiovascular disease [15,16,17,18,19]. Thus, scientists attempt to investigate whether MSCs therapy could also mitigate radiation injury. Here, we will first introduce the underlying mechanisms of radiation injury and the features of MSCs briefly. Then, we focus on the recent progress on MSCs therapy in treating radiation injury. Last, we discuss the challenges and future perspectives of the MSCs therapy.

## 2. Pathophysiological Mechanisms of Radiation Injury

RT utilizes high doses of radioactive energy, known as ionizing radiation (IR), to kill cancer cells. Notably, IR also injuries the healthy cells around the tumor, causing various complications. However, the pathophysiological mechanisms of radiation injury remain mostly unclear. IR induces increased production of reactive oxygen species (ROS), referred to as oxidative stress, injuring cell components such as DNA, proteins, organelles, etc. [20]. The damages to DNA mainly comprise single- and double-stranded breaks and base lesions [21]. Incorrect DNA repair would give rise to mutagenesis or chromosomal instability resulting in cell apoptosis and carcinogenesis [22]. Excessive ROS activates unfolded protein response in the endoplasmic reticulum (ER), which further elicits Ca^2+^ release from ER, causing ER stress [23]. If the ER stress was uncontrolled, the unfolded protein response pathways trigger downstream signals such as c-Jun N-terminal kinase and Bcl-2 protein family members, initiating cell apoptosis or autophagy [24]. The enhanced ROS and imbalanced Ca^2+^ in the cytoplasm cause mitochondrial membrane permeabilization [25], leading to Bax’s activation and the release of cytochrome c, promoting apoptosis development [26]. Moreover, mutated mitochondrial DNA, impaired PPAR-α pathways, and dysregulated ROS production induce mitochondrial dysfunction [26]. The proper functionality of cellular components is closely connected with the cell fate. Thus, clarifying the alterations of intercellular and intracellular signal cascades is beneficial for understanding the radiation injury.

Inflammatory responses, endothelial cell injuries, and fibrosis are vital radiation injury features [27,28,29]. At the acute phase after IR, inflammatory cytokines (tumor necrosis factor, TNF; interleukin-1, IL-1; IL-6; IL-8), chemokines (C-C motif chemokine ligand, CCL; C-C motif chemokine, CXC), and adhesion molecules (intercellular cell adhesion molecule, vascular cell adhesion molecule, E-selectin) are secreted, inducing vasodilation and vascular permeability [30]. Subsequently, coagulation cascade signals are triggered, and endothelial basement membrane is degraded, enabling clearance of damaged tissue and repairing initiation. This acute response may sustain from minutes to several days after IR [29]. Notably, chronic inflammation and oxidative stress would induce fibrosis at the later phase of diseases [31]. The transforming growth factor-β1 (TGF-β1)/Smad signaling has been recognized as the primary player that mediates myofibroblasts proliferation and regulates extracellular matrix and collagens deposition [32]. IR also upregulates the connecting tissue growth factor levels that can enhance the binding of TGF-β1 with its receptor (Smad2, Smad3), promoting fibroblast trans-differentiation [33]. By dissociating TGF-β from its complex, the enhanced ROS promotes TGF-β1/Smad signaling, which further modulates ROS generation via upregulating NADPH oxidase 4 transcriptional activity [34]. Moreover, myofibroblasts are also found to originate from the process named epithelial or endothelial to mesenchymal transition [35]. Other profibrotic cytokines, such as CCL3, CCL2, IL-1, and IL-6, are also essential for fibrosis progress. Elevated IL-6 levels post IR is correlated with radiation toxicity in breast cancer patients and the degree of fibrosis in the irradiated lung [36,37]. Fibrosis formation is usually a chronic but ongoing progressing process, and it lacks sensitive tools allowing for early detection.

Apart from these mechanisms, telomere erosion, miRNAs alterations, epigenetic regulations, and stem cell damage are also engaged in the pathophysiological development of radiation injury [38,39,40,41]. Moreover, these underlying mechanisms interconnect with each other and vary depending on the tissue/cell types, IR patterns (types, doses, and dose rates), and patient-related factors (individual comorbidities and risk factors, such as body mass index, smoking, and genetic predisposition). Thus, determining factors that promote radiation injury progression from asymptomatic remains challenging.

## 3. Characteristics of MSCs

Currently, there is no absolute definition of MSCs. To facilitate the development of MSCs-based study, the International Society for Cellular Therapy proposes several minimal criteria identifying MSCs [42,43,44]. Firstly, surface CD antigens are the most primary and necessary verification method. MSCs positively express stro-1, CD44, CD73, CD90, and CD105. Different from hematopoietic stem cells, MSCs lack CD34, CD45, CD14 (or CD11b), CD79α (or CD19), and HLA-DR. Secondly, MSCs are considered to be plastic-adherent when cultured under standard conditions. Lastly, MSCs must possess the capability of differentiating into osteoblasts, adipocytes, and chondroblasts. This report largely standardizes the definition of MSCs and instructs investigators to estimate the authenticity of their cells.

MSCs can be obtained from multiple tissues (bone marrow, adipose tissue, peripheral blood, umbilical cord, and placenta), providing researchers with great convenience and increasing its clinical application popularity [45]. MSCs derived from differed tissues show distinct characteristics, including proliferation and differentiation potential, paracrine effect, immunophenotypes, and immunomodulatory capacity [46,47]. For example, umbilical cord blood-derived MSCs (UC-MSCs) show more significant proliferation and slower senescence compared with that from bone marrow (BM-MSCs) and adipose tissue (AT-MSCs) [48]. However, BM- and AT-MSCs are capable of tri-lineage differentiation (osteogenic, adipogenic, and chondrogenic) under respective culture conditions, while placenta- and UC-MSCs only differentiate into two cell lineage [46]. Additionally, discrepant paracrine activity reflected by the expression of various cytokines and growth factors was observed in UC- and AT-MSCs [49]. All these differences may influence the function of MSCs from multiple sources. A comprehensive understanding of these features would promote a more efficient clinical application of MSCs.

In most MSCs-based therapy studies, immunomodulation is regarded as the leading factor of the therapeutic property. MSCs can interact with immune system cells (T cell, B cell, natural killer cells, etc.) and regulate immune response depending on direct cell-cell contact and various immunomodulated factors [50]. High inflammation levels would stimulate MSCs to release anti-inflammatory cytokines, inhibiting overactivated inflammation and immune responses. The involved molecules include inducible nitric oxide synthase (iNOs), TGF-β, IL-10, prostaglandin E2 (PGE2), and hepatocyte growth factor (HGF) [51]. T cells would be deactivated by inducing apoptosis or suppressing proliferation [52]. On the contrary, the silent immune system would induce the pro-inflammatory phenotype of MSCs to ensure basic self-defense against the external pathogen. Such plastic immunomodulation function protect tissue against pathogen invasion or self-attack, making MSCs a popular object in the study of tissue repair and regeneration [53].

## 4. Current Attempts of MSCs for Mitigating Radiation Injury

Considerable progress in medications has dramatically reduced the mortality and morbidity of cancer patients. The increased number of cancer survivors enables clinicians to realize the side effects of related treatments such as RT. To date, it has gained remarkable improvements in achieving high-precision RT. For instance, breast cancer patients receiving IMRT exhibited significantly lower occurrence, severity, and persistent of radiodermatitis than those receiving conventional RT [8]. A significant reduction in gastrointestinal toxicity was observed in IMRT than conventional two-dimensional RT (IMRT vs. RT: 33% vs. 77%) [6]. Moreover, the combination of IGRT and IMRT (IG-IMRT) showed more significant superiority than conventional three-dimensional conformal RT in the treatments of rectal cancer and hepatocellular carcinoma [7,9]. With IG-IMRT, hepatocellular carcinoma patients showed longer median survival (IG-IMRT vs. RT: 44.7 vs. 24.0 months) [7,9]. Although modern RT doses have been minimized and are precise, radiation complications still typically occur acutely or chronically. Here, we mainly discuss the latest advances in MSCs therapy application mitigating radiation injury involving the skin, intestine, brain, lung, liver, and heart.

### 4.1. MSCs in Radiation-Induced Skin Injury

Radiation-induced skin injury or radiodermatitis is the most common side effect in people exposed to IR. Up to 95% of cancer patients undergoing RT experienced radiodermatitis [54]. Among the manifestation of radiodermatitis, erythema is the most apparent and mild symptom (incidence with more than 90%), followed by moist desquamation (incidence of 30%) [55]. These varying severity levels are associated with direct radiation injuries and consequent inflammations affecting different skin structures, including epidermis, dermis, and vasculature (well described in [56,57]). The release of cytokines and chemokines by recruited immune cells activates dermal fibroblasts, causing chronic dermatitis and skin fibrosis [58]. Regular treatment of radiodermatitis comprises self-care (daily hygiene habits, loose clothing, avoiding tobacco and alcohol, adequate water intake, etc.) and prophylactic topical corticosteroids [59]. Such therapies are usually based on hearsay or physician preferences lacking powered studies to demonstrate their efficiency [60,61]. The occurrence of radiodermatitis has destroyed patients’ physical appearance and beauty, and also delayed wound healing [29]. Thus, novel therapeutic validating by a more systematic and rigorous design is urgently needed.

It has demonstrated that bone marrow-derived cells such as MSCs, endothelial progenitors, and myelomonocytic cells are recruited to the injured sites by chemotactic signals SDF-1 and CXCR4 participating in the healing process [62]. The intravenous injection of MSCs significantly accelerates the wound healing rate [63]. Increased survival of BM-MSCs ameliorates injury induced by IR combined with traumatic tissue injury [64]. Thus, scientists have attempted to mitigate radiodermatitis using exogenous administration of MSCs. For instance, Moghaddam et al. intradermally transplanted AT-MSCs (2 × 10^6^) to guinea pigs receiving 60 Gy abdominal radiation. These irradiated guinea pigs showed alleviated skin damage, and the combination of low-intensity ultrasound enhanced the curative effect of AT-MSCs [65]. However, the exact mechanism underlying the therapeutic potential of MSCs for radiodermatitis is unclear. Anti-inflammation and anti-fibrosis may be the main ways for MSCs to inhibit radiation injury [66,67]. Inflammation-related cytokines (IL1β and IL10) were regulated by BM-MSCs (5 × 10^5^) in radiation mice models with a 35 Gy dose [67]. Similarly, BM-MSCs injection (2 × 10^6^) via tail vein efficiently reduced 45 Gy radiation-induced rats’ skin fibrosis reflected by decreased TGF-β1 [66]. Notably, the MSCs conditioned medium (CM) could also accelerate wound healing after pipetting onto the irradiated rats’ skin wound [68]. This result indicated that paracrine factors from MSCs play a critical role in repairing radiodermatitis by mitigating the injury site’s inflammatory microenvironment. Apart from animal studies, limited clinical trials were also carried out. A case report analyzed the treatment potential of cadaveric MSCs on a necrotic ulcer in a patient receiving 50–60 Gy dose RT for right leg angioma [69]. Two years after the treatment, clinicians observed a reduced ulcer size and improved the skin quality, confirming the MSC therapy’s efficiency. Thus, MSCs or their secretiome could be novel therapeutics for mitigating the radiodermatitis.

### 4.2. MSCs in Radiation-Induced Intestinal Injury

Radiation-induced intestinal injury (RIII) or radiation enteropathy develops in RT-treated patients with abdominal or pelvic tumors. About 60–80% of patients have nausea, abdominal pain, and diarrhea within 2–3 weeks of RT [70]. Such symptoms usually disappear within 1–3 months of completing therapy. However, a few patients may experience delayed RIII, including disorders in intestine motility and nutrient absorption. Some severe chronic RIII may progress to intestinal obstruction or perforation and fistulae formation. The pathological changes in acute RIII involve inflammation reaction and consequent crypt cell death [71,72]. On the other hand, chronic RIII is more complex and is characterized by mucosa atrophy, intestinal wall fibrosis, and microvascular sclerosis [70]. Numerous preclinical studies utilizing natural products [73], peptides [72], and small molecules [74] to alleviate RIII have been carried out. However, researcher have not yet reached a consensus on the clinical application. Amifostine, a free-radical scavenger, is the earliest drug proved by the FDA to mitigate radiation therapy-related injury [75]. Nevertheless, the narrow treatment time window and lingering concerns of amifostine hinder its clinical uses [76]. Moreover, the US FDA has approved Neupogen and Neulasta in 2015 and leukine in 2018 for acute radiation syndrome [77]. Thus, novel therapeutic strategies are eagerly needed, especially drugs specific for each radiation-induced organ injury.

MSCs were initially found to migrate and settle in the injured intestine after RT [78]. Lately, studies revealed that the transplanted MSCs can reverse the disrupted intestinal function by RT [79,80]. Such benefits were attributed to the MSCs secretome-mediated intestinal regeneration via inflammation inhibition, neovascularization, and epithelial homeostasis maintenance [81]. Additionally, there exist specific stem cells in the intestinal crypt responsible for intestinal repair and regeneration [82]. BM-MSCs (1 × 10^6^) transplantation via tail vein injection was found to increase Lgr5^+^ intestinal stem cell populations, thus facilitating the repair of radiation-induced intestinal injury via activated Wnt/β-catenin signaling [83]. Based on the excellent paracrine effect, MSCs-CM were also applied to preclinical experiments of RIII. Repeated injection of AT-MSCs-CM (abundant angiogenic factors such as IL-8, angiogenin, HGF, and vascular endothelial growth factor) promoted intra-villi microvascular recovery in the irradiated intestine via activating the PI3K/AKT signal pathway [84]. Nevertheless, MSCs cultured under normal conditions only secrete slight cytokines that may possess unsatisfactory therapeutic potential. Given this, Chen et al. pretreated BM-MSCs with pro-inflammatory factors (TNF-α, IL-1β, nitric oxide) and found an enhanced paracrine effect of MSCs, primarily represented by the secretion of IGF [85]. The pretreated BM-MSCs-CM exhibited a more significant therapeutic efficacy in modulating inflammatory responses and mediating epithelial regeneration [85]. Moreover, other modifications such as carrying foreign genes (HGF, CXCL12) or cytokines (R-Spondin1) and engineered MSCs (hydrogel loaded) have also been tested for their capacity in alleviating RIII [86,87,88]. Preclinical studies have shown the therapeutic potential of MSCs (modified or not) in treating radiation injury. MSCs was also tested for clinical treatment of RIII, in which reduced intestinal inflammation and hemorrhage were exhibited after systematic usage of MSCs [89]. However, a detailed treatment strategy remains unknown.

### 4.3. MSCs in Radiation-Induced Brain Injury

Radiation-induced brain injury (RIBI) is mainly presented as cognitive dysfunction in patients experiencing head and neck RT [90]. The degree of tissue injury is unequal based on different periods (acute, early delayed, late delayed) [91]. Acute response is sporadic under current RT techniques. Early RIBI involves angioedema and manifested clinically as headache and drowsiness [92]. Acute and early RIBI are generally recovered within 1 to 6 months. However, late RIBI often represents severe irreversible lesions such as vascular injury and demyelination, leading to ultimate white matter necrosis and brain atrophy [93,94]. Apart from the vascular endothelial cells, neurons and glial cells are also susceptible to IR [95]. In all, RIBI is intractable due to the complex dynamic process [91]. Early epidemiological data showed 11% of morbidity of severe dementia in cancer patients receiving whole brain radiation [96]. In fact, sensitive neurocognitive tests suggested that 90% of irradiated patients had neurological impairment [97]. With regard to the treatment of RIBI, anti-inflammatory drugs have been applied to counteract RIBI, such as eicosapentaenoic acid and fenofibrate [98,99]. Moreover, traditional Chinese medicines are also beneficial for neuroprotection against radiation [100]. In preclinical studies, intrahippocampal transplantation of human neural stem cells restored neural plasticity of irradiated rats by improving the expression of activity-regulated cytoskeletal [101]. At present, MSCs-based cell transplantation and secretome administration are also considered as therapeutic strategies preclinically. UC-MSCs (1 × 10^6^) transplantation via caudal vein infusion showed anti-inflammatory and anti-apoptotic effects on mice with RIBI [102,103]. The RT-triggered inflammation was inhibited, reflected by the decreased IL-1, TNF-α, and the increased IL-10 [102]. On the other hand, the downregulation of pro-apoptotic proteins (p53, Bax) and the upregulation of anti-apoptotic Bcl-2 confirmed apoptosis reduction. This anti-apoptotic benefit was further enhanced through the combined administration of UC-MSCs and nimodipine [103]. MSCs-mediated regulation of both inflammation and apoptosis rescued neurons and astrocytes from necrosis. Additionally, microglia were activated during RIII and initiated inflammation reaction by cytokine and chemokine secretion [104]. Intensive inflammation further accelerated microglia pyroptosis related to the increased expression of NLRP3 inflammasome and caspase-1 [105]. Human trophoblast-derived MSCs (1 × 10^5^) transplantations via brain cortex are able to reverse the microglia pyroptosis, promoting tissue repair [105]. Others also identified that the intranasally administered human MSCs (5 × 10^5^) restored neurological function by reducing inflammation and oxidative stress via declined damage-induced c-AMP response element-binding signals [106]. Unfortunately, only a few researches on applying MSCs therapy in RIBI have been reported so far. The finding that MSCs are also homed to gliomas would encourage more efforts to be devoted to this area [107].

### 4.4. MSCs in Radiation-Induced Lung Injury

Thoracic tumors patients receiving RT tend to suffer from radiation-induced lung injury (RILI) with a mortality of approximately 15% [108]. The RILI is a complex dynamic process, including early pneumonitis and delayed pulmonary fibrosis [109]. The common pathological changes of RILI include epithelial and endothelial cell injuries, inflammatory responses, resulting in the dysfunction of the blood-air barrier and vascular permeability [109]. Moreover, the alveolar macrophages are also stimulated to secrete abundant cytokines (TGF-β1, TNF-α, IL-1β, IL-6, and IL-12) that further participate in the inflammatory process [110]. TGF-β1 is an essential factor that mediates alveolar epithelial cells undergo EMT, a typical feature of fibrosis [111]. The occurrence of a vicious cycle of inflammation would promote delayed pulmonary fibrosis. Once the fibrosis is formed, it is difficult to reverse and leads to a poor prognosis. Apart from the amifostine, steroids, growth factors (IL-7, IL-11, etc.), antioxidants, and signaling inhibitors have been used to treat RILI, yielding unsatisfactory effects [108]. Thus, clinicians ask for novel and more effective therapeutic approaches.

The potential of treatment with MSCs to mitigate RILI has been evaluated and its underlying mechanisms have been explored. A preclinical study showed that BM-MSCs injected into irradiated mice via tail vein could differentiate into lung epithelial and endothelial cells [112]. They also observed an upregulated IL-10 and downregulated TNF-α and TGF-β in RILI mice [112]. Because excessive inflammation and irreversible fibrosis are the leading causes of RILI, the MSCs-mediated anti-inflammation and anti-fibrosis effects may play a vital role in lung tissue repair and regeneration. Consistently, Hao et al. found that intratracheal transplantation of human UC-MSCs (1 × 10^6^/kg) inhibited canine pulmonary inflammation and fibrosis in beagle dogs induced by radiation through reducing IL-1, TGF-β, and hyaluronic acid [113]. Dong et al. first identified two anti-fibrotic factors, HGF and PGE2, that exhibited increased expression in irradiated rat lung tissue after administration of AT-MSCs [114]. Additionally, radiation-induced lung endothelial dysfunction could be alleviated by MSCs-CM [115]. This perhaps further suggested that the paracrine effect rather than differentiation plays a dominant role in the MSCs therapy. In fact, paracrine-depended secretome and vesicles derived from MSCs have also shown a significant efficacy on RILI [116]. Notably, growing evidence showed that gene-modified MSCs may possess more tremendous therapeutic potential than unmodified MSCs in RILI. For example, human UC-MSCs modified with CXCR4 showed a significant anti-fibrotic effect in irradiated mice [117]. This mainly depended on more accurate homing and colonization that was critical for enhancing targeted therapy of MSCs. Liu et al. injected UC-MSCs expressing decorin (an inhibitor of TGF-β and fibrogenesis) into irradiated mice and observed improved lung inflammation and fibrosis [118]. Additionally, manganese superoxide dismutase (ROS scavenger) modified MSCs also exerted a therapeutic effect on RILI reflected by decreased lung cell apoptosis [119]. In fact, gene-modified MSCs overexpress certain soluble factors, which can protect tissues from radiation injury. The combination of natural MSCs properties and overexpressed beneficial factors consolidates the therapeutic effect of MSCs. Despite abundant preclinical evidence of the beneficial effect of MSCs on RILI, relevant clinical data are incredibly lacking. A report involving 11 patients with RILI confirmed autologous MSCs administration safety, but the actual efficacy could not be assessed [120].

### 4.5. MSCs in Radiation-Induced Hepatic Injury

Radiation-induced hepatic injury (RIHI) presents two different clinical types (classic and non-classic RIHI) reflected by distinct characteristics [121]. Both of them occurred in 36% of patients receiving reirradiation for hepatocellular carcinoma [122]. Classic RIHI is recognized by hepatomegaly, anicteric ascites, and increased abdominal circumference [123]. Patients with classic RIHI show upregulated alkaline phosphatase but normal transaminase and bilirubin levels [124]. The veno-occlusive disease, an essential manifestation of classic RIHI, is described as a complete blockage of the central vein by erythrocytes attached to a dense network of reticulin and collagen fibers [125]. Non-classic RIHI represents an impaired liver function in those patients with chronic hepatic injury, such as viral hepatitis and cirrhosis. Jaundice or significantly elevated serum transaminases levels (five times higher than the standard value) could be used to confirm non-classic RIHI [126]. Transaminases are an important biomarker for assessing the hepatic injury. After irradiation, human or rat MSCs perfusion significantly reduced serum transaminase activity, indicating recovered liver function [127,128]. The mechanism might be apoptosis inhibition due to decreased ROS production and increased secretion of anti-inflammatory IL-10 [127]. In another study, the combined intravenous administration of BM-MSCs (1 × 10^6^) and nigella sativa oil present a similar protective effect on the liver [128]. In addition to inherent medicinal value, nigella sativa oil could enhance MSCs homing in injured liver sites. However, Moubarak et al. found that intravenous MSCs were not grafted to the liver but to the intestine following abdominal irradiation. Improved intestinal damage indirectly corrects liver abnormality via enterohepatic recirculation [129]. Meanwhile, the paracrine mechanism played a more critical role and dominated the protection of MSCs against RIHI without liver engraftment. With increased recognition of the paracrine effect, MSCs-CM was also used to examine paracrine factors’ repair capability to RIHI [130]. In vitro administration of MSCs-CM for culturing sinusoidal endothelial cells increased cell viability and blocked apoptosis. In vivo injection of MSCs-CM into irradiated rat reversed radiation-induced hepatic histopathological changes. Critical nutritional factors responsible for the regeneration potential were unclear, but the mechanism may be related to phosphorylation activation of AKT and ERK. Among all beneficial growth factors secreted from MSCs, hepatocyte growth factor possesses multiple tissue repair abilities, especially liver regeneration. Gene-modified AT-MSCs over-expressing HGF downregulated pro-fibrotic proteins (α-SMA and fibronectin) and showed greater anti-fibrotic potential on the irradiated liver in comparison to unmodified MSCs [131]. Unfortunately, there are still no relevant clinical report to date.

### 4.6. MSCs in Radiation-Induced Heart Injury

Apart from the lung, thoracic irradiation also induces heart injury, namely, radiation-induced heart disease (RIHD). RIHDs, such as myocardial, coronary artery, pericardial, valvular, and conduction system diseases, have been observed in breast cancer and Hodgkin’s lymphoma patients [132,133]. These manifestations had a 50% cumulative incidence during 40 years of follow-up in an epidemiological study [132]. RIHD often involves vascular endothelial dysfunction [134], hypertrophy [135], and fibrosis [136]. The underlying mechanisms of RIHD remain mostly indistinct, but the roles of DNA damage, inflammation, oxidative stress, and epigenetic regulation in RIHD have been well illustrated. For the treatment of RIHD, conventional statins and angiotensin-converting enzyme inhibitors are still the first-chosen drugs clinically. With increasing interest in MSCs regeneration therapy, scientists are paying attention to the application of MSCs in RIHD. Vascular injury is the most common feature of RIHD. BM-MSCs (1 × 10^6^/kg) transplantation via tail vein can attenuate radiation-induced artery inflammation and oxidative stress [137]. The repair effect was attributed to the modulation of a series of cytokines and the differentiation potential of MSCs into endothelial cells facilitating vascular regeneration [138]. Additionally, vascular injury is usually accompanied by myocardial fibrosis and cardiac remodeling. Encouragingly, in a RIHD rat model, BM-MSCs (1.5 × 10^6^) transplantation via caudal vein improved myocardial fibrosis and inflammation, which were related to DNA repair and downregulated PPAR-α, PPAR-γ, TGF-β, IL-6, and IL-8 [139]. As mentioned above, MSCs-CM is beneficial to radiation injury repair owing to the paracrine effect. Chen et al. assessed the therapeutic effect of human UC-MSCs-CM on radiation-induced myocardial fibrosis. They found that irradiated human cardiac fibroblasts cultured with UC-MSCs-CM showed greater viability [140]. Inhibited NF-κB activity decreased expression of several pro-fibrotic cytokines, including TGF-β1, IL-6, and IL-8, followed by mitigated collagen deposition and fibrosis [140]. Meanwhile, changes in oxidation markers (malondialdehyde) and antioxidant enzyme levels reflected reduced oxidative stress [140]. However, specific nutritional factors released by MSCs and involved in myocardial protection from IR were not clarified [140]. Thus far, there are few MSCs therapy attempts to manage RIHDs, and abundant evidence is lacking for proving its efficacy. The data on myocardial regeneration suggest that the MSCs therapy is potentially therapeutic to treat RIHD.

## 5. Challenges and Future Perspectives of MSCs Therapy

Although the MSCs have powerful tissue repair capacity due to their paracrine and immunomodulation activity, huge barriers hinder their clinical application. Here, we will focus on safety and efficacy, the two most concerning aspects.

Currently, the relationship between MSCs and tumor has been attracting increased attention. The tumor consists of many types of cells involving a complex pathological environment. Cancer stem cell (CSC) is a kind of multipotent stem cell with great self-renew and differentiation capability in the tumor tissue. Like normal stem cells in the body, CSC is also indispensable for supporting tumor progression, inducing tumorigenesis, maintaining tumor growth, and promoting metastasis [141]. The tumor involves a chronic inflammatory process that recruits endogenous or exogenous MSCs [142,143]. Homed MSCs promote angiogenesis [144] and interact with CSC enhancing the growth [145] and chemoresistance [146] of CSC. The tumor exploits MSCs’ unique immunosuppression nature, allowing malignant cells to escape recognition and clearance by the immune system [147,148,149]. It is reported that once exposed to the tumor microenvironment, MSCs would be reprogrammed and become “allies” of tumor cells, accelerating tumor progress, and invading surrounding normal tissue [149,150,151]. Interestingly, Chen et al. found the engulfment of stromal cells by cancer cells in human breast tumors, and these engulfing breast cancer cells exhibited gene features of MSCs [152]. However, contradictory outcomes about the cancer-promoting effect of MSCs were presented in other studies [153]. For example, several groups found that co-cultured MSCs inhibited melanoma growth by inducing cell apoptosis [154,155]. Colorectal cancer progression could also be attenuated through the intravenous injection of BM-MSCs (1 × 10^7^) [156]. The bidirectional effects of MSCs on tumor development motivate scientists to ascertain more precise mechanisms underlying MSCs and tumor tissue interaction. Unfortunately, it seems that the pro-tumorigenic effect is dominant due to more substantial preclinical evidence. Therefore, MSCs-based therapy must be performed with great caution in clinics, especially with regards to radiation injury patients with malignancy history.

On the premise that security can be guaranteed, investigators need to seek appropriate protocols by which MSCs therapy remedy would maximize radiation repair efficiency. Many questions need to be discussed, for example, how do we determine the selection of the MSC population considering heterogeneity? In addition, the most effective delivery dose and pattern are required to ensure a high retention rate and therapy efficacy. Indeed, different organizational origins give rise to MSCs heterogeneity reflected by diversities of proliferation and differentiation capability, paracrine potential, and immunomodulatory effect [46,47,48]. Despite the minimal criteria mentioned above, it is difficult to sort out homogeneous MSCs. Apart from shared surface CD antigens, there are no additional markers to identify each type of tissue-derived MSCs [157]. Such heterogeneity can lead to the deviation of actual results from expectation and become a significant obstacle to selecting MSCs for clinical usage [158]. Because of the heterogeneity, each MSC population may have distinct therapeutic effects on the same tissue injury. It is necessary to search for the most potent MSC population for radiation injury of a specific tissue. On the other hand, different laboratories have their respective protocols of MSCs isolation, culture, and expansion procedures, causing MSCs heterogeneity and the following difference in quality. Therefore, MSCs management system should be standardized as much as possible. This can reduce heterogeneity caused by different treating conditions and increased comparability among different research results, thus providing valuable clinical guidance of MSCs application. Apart from heterogeneity, the effective dose range and cell delivery route must be emphasized and discussed. A dose gradient experiment of MSCs therapy in radiation injury models should be carried out to find both safe and efficient dose range [159]. In a study of radiation-induced artery injury, a high dose of BM-MSCs (1 × 10^7^/kg) showed greater therapeutic potential in irradiated mice than a low dose of BM-MSCs (1 × 10^6^/kg) [137]. Additionally, different injection patterns, including whole-body infusion via a vein or local interventional injection, will affect the homing of MSCs to injured sites [160]. Thus far, our understandings of the therapeutic effect of MSCs in mitigating radiation injury and the underlying mechanism are basically from preclinical trials. The transition of MSCs administration from animal to clinical studies still requires lots of effort.

## 6. Conclusions

RT is an indispensable part of clinical cancer treatment, and more than 50% of cancer patients received RT [161]. Though the radiation doses and related radiotoxicity have been remarkably reduced due to modern RT techniques, radiation injury in normal tissue is still a thorny problem affecting patients’ life quality and even survival rate. MSCs have abundant resources, excellent regenerative potential, immunomodulatory features, showing therapeutic potential in mitigating radiation injury in preclinical studies. Moreover, chemical, physical, or pharmaceutical preconditioning greatly enhanced the therapeutic potency of MSCs [162]. The overexpression of desired factors (antioxidation, differentiation, immunomodulation, angiogenesis, anti-apoptotic, and regeneration) targeting the specific disease model represents a novel approach in precision medicine. Because the local harsh environment and death signals cause MSCs to be rarely retained in the transplanted sites, MSCs-secretome or a combination with tissue engineering are emerging as a new trend. Notably, radiation-induced skin and intestine injury are easy to be aware of. Radiotoxicity that developed months or years after RT is challenging to be diagnosed or predicted early. In order to reduce or prevent radiotoxicity, more advanced radiotherapy technologies, such as IMRT and IGRT, need to be created. On the other hand, the application of MSCs as regenerative/repair agents when symptoms are presented or as preventive medicine directly after RT also needs careful consideration. The combination of prevention and regeneration/repair is the key to protect radiotherapy patients. Though there are many obstacles in the clinical application of MSCs, there is already a clinical trial evaluating the efficacy of MSC injections for the treatment of chronic radiotherapy-induced complications (PRISME, NCT02814864). We expect a promising future of MSCs therapy in mitigating radiation injury.

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
