# Peer review of "Mesenchymal Stem Cells for Mitigating Radiotherapy Side Effects"

_cells, 2021, doi:10.3390/cells10020294_

Round 1
Reviewer 1 Report
The present manuscript reviews side effects resulting from radiotherapy and MSCs mitigation. MSCs probably would be the one for ameliorating the radiation-sensitive organ injury including GI, lung, liver, heart and brain. The content presented in this review manuscript is interesting to the readers of this journal. Nevertheless, suggestions and comments are listed below to improve the content.
-- Please high lighten the revised areas for this reviewer to spot them efficiently.
Overall:
- The manuscript requires editing and grammatical corrections. I have cleaned up a lot for you, but you still need to have someone performing thorough edit and error corrections.
- The manuscript skips discussion on certain aspects such as skin-wound healing that has been evident in the literature.
- The manuscript has missed references.
Abstract
1). Lines 19-20, I re-write the sentence for you: Therefore, promising treatment strategies to mitigate radiation injury is in pressing need.
1. Introduction
1) Line 29, Correction: there are “no” efficient methods …
2) Line 31, Replace “they” with “cancer patients”.
3) Lines 50-51, Correction: Last, we discuss challenges and perspectives of “the” MSCs therapy.
2. Pathophysiological mechanisms of radiation injury
1) Line 71, Please insert the following reference after 24.
Kiang and Olabisi, Radiation: a poly-traumatic hit resulting in multi-organ injury. Cell Biosci 2019; 9, 25, doi: 10.1186/s13578-019-0286-y.
2) Line 76:Please insert the above reference after “from minutes to several days after IR”.
3) Lines 93, Please replace “dose rate” with “dose rates”.
3. Characteristics of MSCs
1) Line 100: Please insert the following references after 37.
Kiang JG. Adult mesenchymal stem cells and radiation injury. Health Phys 2016; 111, 198-203, doi: 10.1097/HP.0000000000000459.
Kolf CM, Cho E, Tuan RS. Mesenchymal stromal cells. Biology of adult mesenchymal stem cells: regulation of niche, self- renewal and differentiation. Arthritis Res Ther 9:204; 2007.
2) Line 101: Please insert “stro-1, CD44,” before CD73 …
4. Current attempts of MSCs for mitigating radiation injury
1) Line 134: Please replace “Chronic”with “chronically”.
2) Line 138: Please insert “the” between in and radiation-induced …
3) Line 139: Please insert “The” before radiation-induced intestinal …
4) Line 144: Please replace “of acute RIII” with “in acute RIII”.
5) Line 145: Please insert the following reference after 48.
Kiang JG, Smith JT, Cannon G, Anderson MN, Ho C, Zhai M, Cui W, Xiao M. Ghrelin, a novel therapy, corrects cytokine and NF-κB-AKT-MAPK network and mitigates intestinal injury induced by combined radiation and skin-wound trauma. Cell Biosci 2020; 10, 63, doi: 10.1186/s13578-020-00425-z.
6) Lines 146-148: More references need to be included for this statement.
Numerous preclinical studies utilizing natural products (ref), peptides (ref), and small molecules (ref) to alleviate RIII have been carried out.
For peptides (such as ghrelin), the following reference can be cited.
Kiang JG, Smith JT, Cannon G, Anderson MN, Ho C, Zhai M, Cui W, Xiao M. Ghrelin, a novel therapy, corrects cytokine and NF-κB-AKT-MAPK network and mitigates intestinal injury induced by combined radiation and skin-wound trauma. Cell Biosci 2020; 10, 63, doi: 10.1186/s13578-020-00425-z.
For small molecules (such as exosomes and miR-124), the following reference can be cited.
Leavitt RJ, Limoli CL, Baulch JE. miRNA-based therapeutic potential of stem cell-derived extracellular vesicles: a safe cell-free treatment to ameliorate radiation-induced brain injury. Int J Radiat Biol. 2019;95(4):427-435.
7) Lines 148-151: These statements are not correct. The US FDA has approved Neupogen and Neulasta in 2016 and leukine in 2018 for ARS. Amifostine is not the only one any more. You need to revise these statements. One thing is very sure. That is, we need drugs specific for each radiation-induced organ injury.
8) Line 173: Please replace “MSC” with “MSCs”.
9) Line 174: Please insert “the” between in and radiation-induced …
10) Line 175: Please insert “The” before radiation-induced brain …
11) In this subsection (lines 175-201), you have missed articles from Charlie Limoli’s laboratory at UC Irvine. This lab has published a series of articles on this subject. For example:
Acharya MM, Rosi S, Jopson T, et al. Human neural stem cell transplantation provides long-term restoration of neuronal plasticity in the irradiated hippocampus. Cell Transplant. 2015;24(4):691-702.
12) Line 199: Please replace “It is a pity” with “Unfortunately”.
13) Line 202: Please insert “the” between in and radiation-induced lung …
14) Lines 215-216: The sentence is awkward. I re-write it to be “The potential of treatment with MSCs to mitigate RILI has been evaluated and its underlying mechanisms have been explored.”.
15) Line 223: Please replace “fibrotic factors that HCG and PGE2 exhibited” with “fibrotic factors, HCG and PGE2, that exhibited”.
16) Line 224: Please insert “the” between Also and radiation-induced…
17) Line 226: Please replace “a dominant part in MSCs therapy.” With “a dominant role in the MSCs therapy.”
18) Line 227: Please insert “a” between shown and significant.
19) Line 229: Please clarify “single MSCs”. It is confusing.
20) Line 236: Please clarify “protective to irradiate tissues”. It is confusing.
21) Lines 237-238: Because of the side effects resulting from the MSC therapy, the current attention actually is on using MSC-derived exosomes along with miRNAs and anti-inflammatory cytokines to treat the radiation injury. Therefore, you should modify your statements here.
22) Line 242: Please insert “the” between in and radiation-induced …
23) Line 243: Please insert “The” before radiation-induced hepatic …
24) Line 250: Please replace “level” with “levels”.
25) Line 279: Please delete “more”.
26) Line 272: Please insert “the” between in and radiation-induced…
27) Lines 273-274: The sentence is confusing. I made the changes as following.
Apart from the lung, thoracic irradiation also induces heart injury, namely, radiation-induced heart disease (RIHD).
28) Line 275: Please replace “in breast, Hodgkin’s” with “in either breast or Hodgkin’s”.
29) Line 276: Please insert “and” before fibrosis.
30) Line 278: Please insert “and” before epigenetic”.
31) Line 282: Please insert “the” before radiation-induced…
32) Line 289: Please insert “the” before radiation-.
33) Line 290: Please replace “fibroblast” with “fibroblasts”.
34) Lines 296-298: The statement of “The excellent therapy benefits of the MSCs therapy on myocardial regeneration should have a bright future for application in RIHD” does not make sense. I think you try to say “The data on myocardial regeneration suggest that the MSCs therapy is potentially therapeutic to treat RIHD.”
35) You should include a subsection 4.6. MSCs in the radiation-induced skin injury
Radiation delays skin-wound healing (Kiang and Olabisi, 2019). It is reported that the intravenous injection of MSCs significantly accelerates the wound healing rate (Kiang and Gorbunov, 2015).
Kiang JG, Gorbunov NV. Bone marrow mesenchymal stem cells increases survival after ionizing irradiation combined with wound trauma: Characterization and therapy. J Cell Sci Ther2014; 5, 190, doi: 10.4172/2157-7013.1000190.
5. Challenges and future perspectives of MSCs therapy
1) Line 322: Please insert “the” before BM-MSCs…
2) Line 323: Please replace “force” with “motivate”.
3) Line 324: Please replace “Unfortunately” with “However”.
4) Line 328: Please replace ”exert the most excellent” with “maximize”.
5) Lines 329-331: These are not complete sentences. Please re-write them.
6) Line 338: Please replace “MSCs” with “MSC”.
7) Line 339: Please replace “MSCs” with “MSC”.
8) Line 353: Please replace “study to clinical trial” with “studies to clinical trials”.
6. Conclusions
1) Line 356: Please insert reference(s) after RI.
2) Line 360: Please insert “and” before anti-fibrotic.
3) Line 363: Please insert references after exosomes.
4) Line 369: Please replace “outstanding” with “distinct”.
-- Again, please highlight the revised areas for this reviewer to spot them efficiently for reviewing your next revised version.
Reviewer 2 Report
Authors are asked to give more information supported by literture on:
1) administration routes and effective dose of MSCs to be used after radiotherapy treatmens for the different considered pathologies.
2) information on chemical, physical or pharmaceutical MSCs preconditioning is reported. This procedures are relevant in the personalized medicine direction;
3) difference betwenn rigenerative medicine and prevention
4) detailed description of risk analysis
Reviewer 3 Report
Authors should quote others strategies carried out to minimize radiation- induced injuries such as IMRT or IGRT techniques
Round 2
Reviewer 1 Report
-- Please high lighten the revised areas for this reviewer to spot them efficiently.
Overall:
The revised manuscript has been greatly improved. However, some minor revisions are still required.
- page 4, line 179: Insert “the” before treatment… Also insert “the “ before skin quality.
- page 4, line 180: Replace “its” with “their”.
- page 4, line 181: Insert “the” radiodermatitis.
- page 5, line 236: Insert “studies” after preclinical.
- page 9, line 402: Replace “model” with “models”.
- page 9, line 403: Insert “the” before radiation-induced…
- page 9, line 408: Replace “trials” with “studies”.
- page 9, line 419: Insert “the” before specific …
- page 9, line 421: Insert “the” before radiation-.
- page 9, line 423: Replace “to diagnose or predict early” with “to be diagnosed or predicted early”.
- page 9, line 430: Replace “KX. W., L.L., and TS. L.,” with “K-XW, LL, and T-SL”…
Author Response
Thanks for your kind suggestions.
I have revised the manuscript accordingly and high lighten all the revised areas with green color.
Reviewer 2 Report
No comments or suggestions
Author Response
Thanks for your review.
Round 3
Reviewer 1 Report
Accept.